# Machine learning in dentistry: a scoping review

Shrey Lakhotia[1], Hormazd Godrej[2], Amandeep Kaur[3], Chaitanya Sai Nutakki[4], Michelle Mun[5,6☯], Pascal Eber[7,8,9☯], Leo Anthony Celi[10,11,12¤*]

1 Helios Enter Data Warehouse IT Exp., Henry Ford Health System, Detroit, Michigan, United States of America, 2 Independent Researcher, Mumbai, India, 3 Department of Oral Health Sciences, Post Graduate Institute of Medical Education and Research, Chandigarh, India, 4 Department of Computer Science and Engineering, SRM University, Mangalagiri, India, 5 Faculty of Medicine, Dentistry and Health Sciences, Melbourne Dental School, The University of Melbourne, Melbourne, Victoria, Australia, 6 Centre for Digital Transformation of Health, Faculty of Medicine, Dentistry and Health Sciences, The University of Melbourne, Melbourne, Victoria, Australia, 7 Division of Oral and Maxillofacial Surgery, Massachusetts General Hospital, Boston, Massachusetts, United States of America, 8 Department of Oral and Maxillofacial Surgery, Harvard School of Dental Medicine, Boston, Massachusetts, United States of America, 9 Division of Oral and Maxillofacial Surgery, Medical University Hannover, Hannover, Germany, 10 Laboratory for Computational Physiology, Massachusetts Institute of Technology, Cambridge, Massachusetts, United States of America, 11 Division of Pulmonary, Critical Care and Sleep Medicine, Beth Israel Deaconess Medical Center, Boston, Massachusetts, United States of America, 12 Department of Biostatistics, Harvard T.H. Chan School of Public Health, Boston, Massachusetts, United States of America

☯ These authors contributed equally to this work.
¤ Senior author.
* lceli@mit.edu

**Data availability statement:** All data analyzed during this study were obtained from publicly available databases, including PubMed. The

## Abstract

Artificial intelligence (AI), specifically machine learning (ML), is increasingly applied in decision-making for dental diagnosis, prognosis, and treatment. However, the methodological completeness of published models has not been rigorously assessed. We performed a scoping review of PubMed-indexed articles (English, 1 January 2018–31 December 2023) that used ML in any dental specialty. Each study was evaluated with the TRIPOD + AI rubric for key reporting elements such as data preprocessing, model validation, and clinical performance. Out of 1,506 identified studies, 280 met the inclusion criteria. Oral and maxillofacial radiology (27.5%), oral and maxillofacial surgery (15.0%), and general dentistry (14.3%) were the most represented specialties. Sixty-four studies (22.9%) lacked comparison with a clinical reference standard or existing model performing the same task. Most models focused on classification (59.6%), whereas generative applications were relatively rare (1.4%). Key gaps included limited assessment of model bias, poor outlier reporting, scarce calibration evaluation, low reproducibility, and restricted data access. ML could transform dental care, but robust calibration assessment and equity evaluation are critical for real-world adoption. Future research should prioritize error explainability, outlier reporting, reproducibility, fairness, and prospective validation.

search strategy and inclusion criteria are outlined in the manuscript, and a detailed search strategy is provided in the Supporting Information. The extracted data used for this analysis are provided as supplementary material S2 File to ensure transparency and reproducibility. No additional data were generated or analyzed. For further inquiries, please contact the corresponding author.

**Funding:** The author(s) received no specific funding for this work.

**Competing interests:** Leo Anthony Celi is the Editor-in Chief of PLOS Digital Health.

## Author summary

Machine learning (ML) techniques are increasingly applied to imaging-driven clinical specialties such as dentistry. We reviewed all English-language PubMed studies (2018–2023) that applied ML in dentistry. Each paper was evaluated for key reporting areas such as data preprocessing, model validation, clinical performance, calibration, reproducibility, and equity considerations. Among the 280 eligible studies, most of the studies were in the subspecialty of oral and maxillofacial radiology. However, fewer than one-third reported calibration, outlier handling, equity considerations, and reproducibility. We underline the need to address equity to ensure safe implementation of ML in diverse populations. Open-source code and deidentified data will strengthen reproducibility and accelerate innovation. We advocate standardized evaluation criteria to guide responsible ML integration into dental diagnostics and treatment planning.

## Introduction

The use of artificial intelligence (AI) in healthcare has changed how diagnosis and treatments are done across many medical fields [1], providing unique opportunities for making decisions based on data. Dentistry, a field that heavily relies on imaging data, stands to benefit substantially from such advancements. Machine learning (ML), a part of AI, has become an important tool allowing predictive modeling and decision support systems which can improve the accuracy of diagnoses as well as tailor treatment plans to individual patients.

Despite this progress, the use of ML in dentistry is still inconsistent. Some specific areas, such as oral and maxillofacial radiology [2], have shown significant improvement by using ML for tasks limited to image segmentation and disease classification. Also, issues about generalizability, explainability of errors in clinical performance, and fairness of these models haven't been properly dealt with, which makes them difficult to be accepted clinically. These limitations highlight the need for a full analysis of ML models in dentistry.

For example, Arsiwala-Scheppach et al. reviewed 168 papers published from January 2015 through May 2021 and highlighted reporting deficits but did not evaluate calibration performance or fairness [3]. A recent systematic review by Kukreja examined dental-AI adoption in education and clinical practice (7 papers) omitted calibration performance and fairness metrics [4]. By screening the literature from 2018 to 2023, extracting 280 studies across all dental specialties, and systematically recording granular elements of model reporting that include metrics for calibration performance, clinical performance, and fairness, we believe our review provides the first field-wide map of model reliability and equity.

To our knowledge, this is the first review to systematically address these critical aspects to improve the transparency and reliability of ML models in dental practice within the PubMed-indexed literature. This review provides the largest scoping synthesis of ML in dentistry, with 280 studies included over a 5-year period. Through this paper, we seek to advance the responsible and effective adoption of ML in dentistry, ensuring that it meets the demands of diverse clinical settings and patient populations. The research question addressed in this scoping review is: What applications of machine learning have been reported in dentistry, how are model performance metrics documented, and to what extent do published studies address equity and reproducibility?

## Materials and methods

The review was conducted as a scoping review following the Arksey & O'Malley framework (as updated by JBI). Reporting adhered to the PRISMA-ScR (Preferred Reporting Items for Systematic reviews and Meta-Analyses extension for Scoping Reviews) guidelines [5]. This ensures transparency and rigor in reporting scoping reviews. The PRISMA-ScR checklist has been completed and is provided as supplementary material S1 File. Before the evaluation started, the protocol was not included in a public database such as Open Science Framework (OSF) due to the exploratory nature of the review.

### Scope of the review

This scoping review evaluates ML models applied in dentistry (2018 – 2023) using the PICO framework [6].

### Information sources and search strategy

We queried PubMed using Boolean operators to combine search terms effectively. PubMed was selected for its broad coverage of topics in dentistry and machine learning, ensuring clinical relevance of included studies. Boolean operators such as OR were used to expand the scope by including synonyms or related terms and AND was used to combine the terms related to machine learning and dentistry simultaneously. We started with the search terms: "machine learning" OR "deep learning" AND "dentistry". After evaluating the preliminary results, which involved consultation among reviewers with expertise in both dentistry and data science, additional terms such as - "neural network", "artificially intelligent", "natural language processing", "algorithm", "Artificial Intelligence", "stomatology" and "oral medicine" were included to expand the scope of the search. The final query was applied as a single combined search. S1 Table provides the full search query for clarity and reproducibility.

The most recent search was conducted on December 27, 2024.

### Selection process

The search was performed with Boolean operators, and only original studies published in print and in English between 2018 and 2023 were considered. We employed a two-stage review methodology to review the resultant group of 1506 studies. Stage one involved the initial screening of identified studies. Reviewer teams, each consisting of a data scientist and a dentist, independently assessed titles, abstracts, and, if needed, a full-text review to determine eligibility based on the inclusion criteria. Any discrepancies within a team were adjudicated by a third reviewer (dentist). Stage two focused on data extraction from the studies confirmed for inclusion after the initial screening.

### Data collection process

The nbib file, a standard bibliographic file format that contains metadata such as titles, authors, publication dates, and abstracts, was downloaded from PubMed from the search results. This was imported into Zotero [7], a reference management tool, to organize the bibliographic information. A csv file was exported from Zotero that had the relevant fields like titles, summaries, the year when published, as well as authors, and uploaded onto Google Sheets. Extra columns in this sheet were added to include all the criteria used in the evaluation based on the Tripod+AI guidelines [8]. In stage one, answers were recorded for each paper individually by 2 reviewers regarding the paper meeting inclusion criteria as

categorical- yes/no/maybe responses. Any paper that was marked maybe or where the 2 reviewers' responses didn't match was reviewed by a third reviewer and the final response was recorded. In stage two, all the papers that had "Yes" responses in stage one, were included. For this stage, reviewer pairs (each comprising a dentist and a data scientist) reviewed the full-text articles collaboratively and reached a consensus before systematically logging the required data points. In case of any clarification needed during this process, other reviewers were engaged. Data was logged by marking most criteria with binary yes/no responses, and categorizing certain criteria with descriptive categorical variables from the included studies as defined in the S2 Table. For some studies that we could not find complete papers from public sources, we got access through researchers linked to educational institutions. This study was based solely on the analysis of published studies and did not involve human participants or patient-level data. Therefore, ethical approval was not applicable.

## Eligibility criteria

This review followed the Arksey & O'Malley framework (as updated by JBI) and integrates the PICO elements directly into the inclusion criteria below.

**Inclusion criteria:** PICO stands for Population, Intervention, Comparator, and Outcome, and was used to structure the review. Studies were included if they met all the following conditions:

1. **Population (P)** – Original research involving human dental patients.
2. **Intervention (I)** – Studies that propose a machine learning model using individual patient-level data intended for direct point-of-care use that performs any of the following tasks (a task is considered as the main objective of the ML model addressed in the paper):
   - Classification (predicts categorical outcomes)
   - Regression (predicts a continuous outcome)
   - Segmentation (predicts the specific region or structure in an image)
   - Generation (generation of new data)
   For the purpose of this review, 'direct point-of-care use' was interpreted as models designed to directly inform the clinician of an immediate clinical decision, such as diagnostic (e.g., identifying caries on a radiograph), prognostic assessment (e.g., predicting treatment success), or treatment planning for an individual patient. This excluded models developed primarily for administrative purposes, population-level risk stratification, or image processing tasks (e.g., image quality enhancement) that did not themselves yield a diagnostic or prognostic output, even if they might indirectly support clinical decisions.
3. English language, published in print 2018 – 2023, full text accessible. We chose this time period to capture the trends in this topic over a longer time frame, given the growing interest in the last couple of years.
4. **Study Type** – Only peer-reviewed, original research articles reporting primary data were considered eligible; reviews, editorials, commentaries, and similar publication types were excluded.

The Comparison elements (C) and the Outcome (O) of the PICO framework were descriptively treated: the studies were eligible regardless of whether they reported a comparator (e.g., clinician evaluation) or outcome (e.g., specific performance metrics). Whenever such data were present, we extracted them for synthesis.

**Exclusion criteria:** Studies were excluded if they met any of the following:

1. The described "algorithms" were **standards based on expert judgments**, not derived from data using machine learning.
2. The paper was **inaccessible** despite institutional access (e.g., two articles published in 2022 and 2023 were omitted for this reason).
3. The paper was a preprint.
4. The study focused on **data processing tasks** (e.g., data optimization) without developing an ML model.

In total, we chose 280 papers to examine and assess by applying the Tripod+AI guidelines to ensure transparency and reproducibility of the research.

The study selection process is summarized in Fig 1.

Each investigation was evaluated based on predetermined criteria to ensure consistency and rigor. The evaluation process focused on the following key elements:

- **Clinical goal**: The clinical outcome targeted by the machine learning model, as defined in the abstract or introduction.
- **Dental specialty**: The dental specialty addressed by the model (e.g., orthodontics, oral radiology).
- **Model type**: The type of machine learning model used (e.g., classification, regression, segmentation).
- **Outlier reporting**: Whether the study reported handling of outliers.
- **Validation strategy**: Whether the study used any approach to validate the model's performance, including k-fold cross-validation or external datasets.
- **Performance metrics**: Evaluation of model performance, including discrimination (e.g., AUROC), calibration (e.g., Brier score), and clinically relevant metrics (e.g., PPV, NPV).

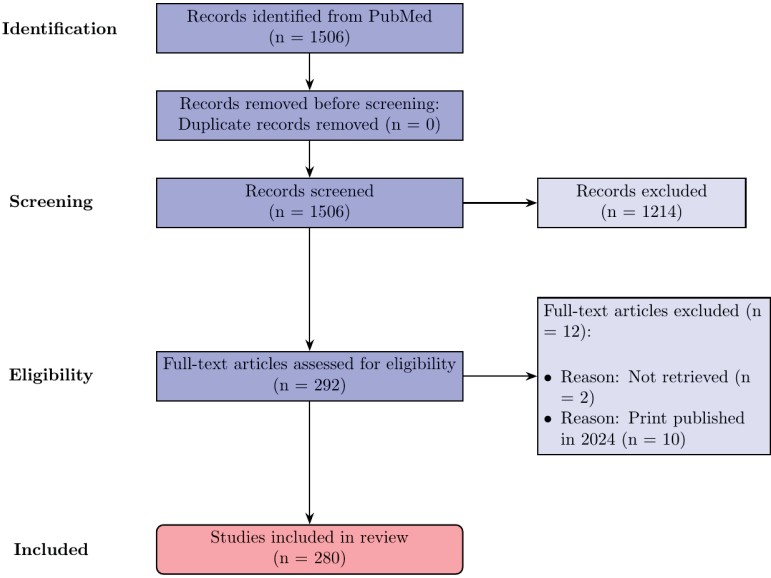

**Fig 1. PRISMA flow diagram for study selection.**

- **Interpretability and bias**: Whether the study reported methods to interpret model outputs (e.g., SHAP values) and addressed bias or fairness.

The full list of evaluation criteria, along with their detailed descriptions and definitions, is provided in S2 Table.

## Synthesis of results

The extracted variables will be summarized as percentages.

Subgroup analyses will be performed based on: (a) dental specialty (e.g., orthodontics, oral radiology), (b) model type (e.g., classification, segmentation), and (c) machine learning approach (supervised vs. unsupervised).

We will present the synthesis in three ways: (a) a brief narrative summary highlighting key findings; (b) a summary table listing the percentage of key features (e.g., validation strategy, calibration reporting, bias assessment); and (c) a stacked bar chart that visualizes the distribution of studies by publication year and dental specialty.

# Results

## Overall study characteristics

The overall study characteristics of the 280 included studies are as below:

**Study design**:

- **267 (95.36%)** of studies were retrospective.
- **10 (3.57%)** of studies were prospective and **3 (1.07%)** were both retrospective and prospective.

**Data handling**:

- **236 (84.29%)** of studies did not report any handling of outliers.

**Model tasks**:

- Classification tasks were the most common, conducted in **167 (59.64%)** of studies.
- Generative tasks appeared in **4 (1.43%)** of studies, beginning from 2021 onward.

**Dental specialties**:

- Oral and maxillofacial radiology comprised **77 (27.50%)** of studies, making it the most researched specialty.
- Endodontics, pediatric dentistry, oral medicine, orofacial pain and dental anesthesiology were the least represented, together accounting for **8.93%**.

## Measures of model performance

- **Validation Strategies**: **14.64%** of the studies did not report any validation strategy.
- **Calibration metrics**: Calibration metrics, such as Brier scores and calibration plots, were reported in only **20.36%** of studies, indicating a lack of focus on the accuracy of predicted probabilities.

- **Clinically relevant metrics:** Among the included studies, 68.21% reported at least one or more clinically relevant performance metrics, such as true positives, true negatives, false positives, and false negatives. However, as detailed in the Discussion, a comprehensive discussion of the clinical implications of these specific error types was often lacking in the studies.

### Bias and fairness considerations:

Only **25.71%** of studies explicitly addressed bias, fairness, or generalizability to diverse populations, underscoring an area needing significant improvement.

### Subgroup analysis results

1. **By dental specialties (**considering specialties with 10 or more papers):
   (a) Studies related to endodontics focused primarily on classification tasks (**80%**).
   (b) Oral and maxillofacial surgery had the highest percentage of regression tasks (**26.19%**).
2. **By model tasks**:
   (a) Classification tasks dominated, accounting for **59.64%** of studies.
   (b) Segmentation, regression, and generative tasks followed, comprising **23.21%**, **15.71%**, and **1.43%**, respectively.
3. **By machine learning approach**:
   (a) Supervised approach was the most common (**98.21%**).

### Key findings

- Discrimination and calibration performance evaluation: There is a notable discrepancy in the number of studies reporting discrimination performance (86.43%) compared to reporting calibration performance metrics (20.36%). This issue brings up questions regarding the trustworthiness of these models and the associated systemic risks when used within clinical setups.
- While 68.21% reported clinically relevant performance metrics, there was limited discussion of the distinctions and implications of such metrics. These metrics are essential for dental professionals to evaluate the practical utility of adopting these machine learning (ML) models in real-world clinical workflows. For instance, understanding how false negatives (missed diagnoses) differ from false positives (overdiagnoses) could provide insights into understanding the models' reliability in clinical practice.
- Outliers: 84.29% of studies did not mention outlier removal. This is critical as improper handling of outliers can lead to bias in the model parameters.
- Reproducibility: Methodologies were described in a manner that allows reproduction in 30.71% of the studies. Likewise, datasets are available to the public in 11.79%.
- Bias and equity: Just 25.71% of research took into account matters tied to bias and fairness. This emphasizes the potential threat of propagation of health inequality when the models are used in real-life situations.

These findings are summarized in Table 1.

See Fig 2 for a stacked bar chart showing the annual number of ML studies by dental specialty.

The results highlight the urgent need for better standards in assessing machine learning models, especially to equitable healthcare outcomes for every patient demographic group and more reproducibility.

**Table 1. Key findings: performance metrics and bias reporting**

| Metric | Percentage (%) |
|---|---|
| Outliers reported | 15.71 |
| Discrimination metrics reported | 86.43 |
| Calibration metrics reported | 20.36 |
| Clinically relevant metrics reported (e.g., true positives, true negatives, false positives, false negatives) | 68.21 |
| Interpretability tools used | 39.64 |
| Bias/fairness addressed | 25.71 |
| Datasets publicly available | 11.79 |

Notes: The table summarizes the key findings regarding the performance metrics and bias reporting in the reviewed studies.

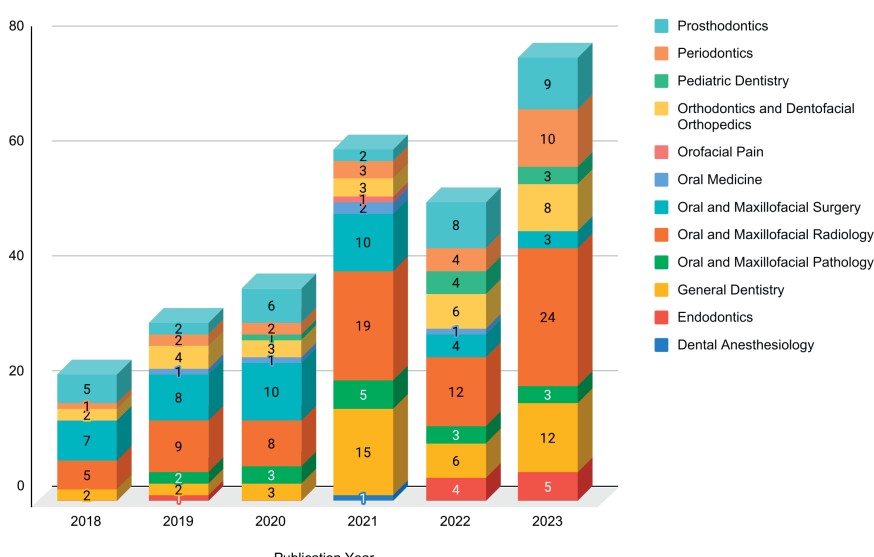

**Fig 2. Annual count of studies by dental specialty.** Stacked bar chart showing the number of machine learning papers published per calendar year from 2018 through 2023 (n = 280). Each bar is subdivided by dental specialty (colour legend at right).

The study underscores the necessity for consistent reporting criteria in future research by highlighting the variation in methodological quality among the included studies.

## Discussion

In this scoping review, we evaluated the papers proposing machine learning (ML) models in dentistry, focusing on different criteria related to study design, data management, methods, measures of model performance, and consideration of bias and fairness. Our results show that there is a nascent interest in unsupervised approaches in machine learning (ML) research in dentistry since 2021.

To ensure the efficient and fair implementation of ML in dentistry, several issues must be resolved:

### More ML research in other dental specialties

The potential of ML in diagnostic imaging and treatment planning is highlighted by the predominance of studies in oral and maxillofacial radiology (27.50%), oral and maxillofacial surgery (15.00%), and general dentistry (14.29%). Methods including random forests, support vector machines (SVMs), and deep learning have been used to improve diagnostic precision and forecast treatment results. Notwithstanding these developments, the sparse use of ML in other dental specialties suggests a need for broader exploration and implementation.

### Outlier reporting

Outlier removal in medical imaging is a nuanced topic that needs to be implemented to reduce model bias and improve performance metrics. As compared to numerical data, implementing this in imaging data will require a specialized approach.

### Model performance metrics

The majority of the studies have reported discrimination performance metrics, but far fewer reported calibration metrics. In high-risk environments like healthcare, the models published should address calibration, as models need to be reliable in quantifying risks to ensure transparency, trust, and equity in the use of AI in healthcare. When ML models are not calibrated, they may produce overconfident predictions that skew clinical judgments and may misallocate resources. For example, a poorly calibrated model may exaggerate risk, resulting in needless treatments, or underestimate illness risk in specific populations, resulting in insufficient therapy.

### Interpretability and explainability of errors

In our study, at least one measure of a clinically important performance indicator, such as true positives, true negatives, false positives, and false negatives, was reported in 68.21% of studies. These metrics are pivotal in assessing model performance beyond general accuracy or other broad measures.

Although interpretability is essential for establishing trust in machine learning (ML) models, focusing just on it runs the risk of reinforcing bias by oversimplifying each decision an ML model makes and neglecting the underlying systemic biases in the data. Therefore, we advocate for more emphasis on the explainability of errors in ML model performance rather than on their general interpretability. For example, false positives (overdiagnoses) might result in wasteful procedures, while false negatives (missed diagnoses) can postpone critical treatments. Similarly, true positives and true negatives provide insights into the model's strength in correctly identifying clinical outcomes.

Researchers can increase clinician trust, enhance patient outcomes, and guarantee a safer integration of machine learning into dental practices by highlighting the explainability of these errors.

### Equity, bias, and fairness considerations

Few studies addressed issues of bias, fairness, or generalizability across diverse populations. This is critical, as the datasets used for training ML models, more often than not, do not capture the implicit variables that affect the outcomes across geographies and at different times. Hence, such models may not perform adequately in varied demographic settings, potentially exacerbating health inequity. Obermeyer et al. [9] demonstrated the importance of subgroup

analyses by showing that a widely used clinical model systematically underestimated health risk among Black patients. Their findings underscore the critical importance of selecting appropriate outcome variables: the original model predicted healthcare costs, which did not accurately reflect true health status due to underlying systemic inequities in healthcare access.

## Regulatory and ethical implications

Following legal and ethical guidelines is essential when integrating machine learning into healthcare practice. In its proposal for an artificial intelligence law, the European Commission highlights the importance of trustworthy AI by classifying AI applications according to risk categories and defining specifications for high-risk systems [10]. Dental machine learning applications, especially those that impact clinical decisions, can be classified as high-risk, requiring strict adherence to guidelines to ensure patient safety [11]. With the recent increase in FDA-approved medical devices, it becomes more imperative to recalibrate evaluation standards regularly [12].

## Limitations and future directions

This scoping review has a few limitations. First, this review was limited only to PubMed, which may have excluded relevant studies from other databases. This single-database approach means our findings, particularly claims regarding the 'field-wide' scope or being the 'largest' synthesis, should be interpreted as specific to the PubMed-indexed literature. Second, our reliance on studies published in English may have introduced language bias. Third, our review does not include studies published from January 1, 2024, as the review of the existing 280 studies required considerable time and effort. The inclusion of those published studies may have strengthened the review, as the rapid pace of ML research may render some findings obsolete. Fourth, this review is limited by the heterogeneity of included studies, as the studies differed on the model task and the type of data on which it was trained to perform (e.g., different types of radiographs, photographs, clinical notes, etc.).

Our findings are consistent with previous reviews that highlight the growing use of deep learning in medical imaging, but also indicate shortcomings in bias reporting [13] and calibration performance evaluation [14]. Our specific recommendations for future dental ML research are four-fold. First, bias mitigation strategies such as (a) subgroup analysis; (b) fairness-aware machine learning techniques; and (c) prospective studies with diverse populations should be implemented to promote fairness in dental AI research. Secondly, to enable wider clinical adoption and guide clinical decision-making, it is imperative to include calibration performance metrics such as: (a) reliability diagrams plotting predicted probabilities against actual outcomes, and the calibration error reporting the difference between these two; (b) bin-based error aggregates like expected calibration error (ECE), maximum calibration error (MCE); and (c) scalar proper scoring rules such as Brier score, calibration slope, and intercept. Thirdly, future reviews should incorporate additional databases such as those strong in computer science (e.g., IEEE Xplore, Scopus) apart from PubMed, which is primarily focused on biomedical and health sciences literature. Finally, techniques such as thresholding techniques based on autoencoder neural networks and clustering algorithms can be used to identify image outliers. Future studies should report handling of outliers, even if there were no outliers identified.

Collaborations between dental professionals, data scientists, and regulatory bodies [15] are essential to harness the full potential of ML in dentistry while safeguarding ethical and equitable care. Future reviews should include additional databases to ensure comprehensive coverage.

Critical-care datathons built on the MIMIC (Medical Information Mart for Intensive Care) database [16] that aim to address specific clinical challenges using AI demonstrate how open data, an official scoring script, and a public leaderboard can accelerate research progress. Dentistry currently lacks such community benchmarks. Professional societies and researchers should coordinate to release de-identified dental image repositories with standard train/validation/test splits and automated evaluation scripts, akin to the CheXpert [17] or PhysioNet [18] Challenge frameworks.

### Implications for stakeholders

**Practitioners:** Evaluate calibration plots and subgroup errors before deployment. **Educators:** Incorporate fairness toolkits into dental curricula. **Policymakers:** Require public reporting of calibration and bias analyses for AI devices. **Researchers & professional societies:** Establish open, standardized data repositories (e.g., multi-institutional radiograph sets with de-identified metadata) and public benchmark tasks so models can be compared on identical test sets.

### Conclusion

In conclusion, the application of ML in dentistry has evolved significantly with rapid advancement in the field. However, critical gaps remain in the research, including the evaluation of calibration performance, the reporting of bias and outlier handling, and data and code sharing. Addressing these gaps is needed to avoid reinforcing biases and to ensure equitable patient-centered care.

### Supporting information

**S1 File. PRISMA-ScR checklist.** The completed PRISMA-ScR checklist is provided in the supporting information file labeled "S1_File.pdf" to ensure transparency and adherence to reporting standards.
(PDF)

**S2 File. Data extracted from analyzed studies** This file is provided in the supporting information file labeled "S2_File.xlsx". It contains the data extracted from all included studies.
(XLSX)

**S1 Table. Search strategy for PubMed.** The search strategy is provided in the supporting information file labeled "S1_Table.pdf".
(PDF)

**S2 Table. Evaluation criteria for included studies.** The evaluation criteria are provided in the supporting information file labeled "S2_Table.pdf".
(PDF)

### Acknowledgments

We would like to thank Stuti Agrawal, Dr. Kshitij Chavan, Dr. Rounak Dey, Soaad Hossain, Dr. Kokila Jaiswal, Dr. Gayathri Ramasamy, N P V S Subrahmanya Sastry, Dev Sharma, Professor Ashlesha Shimpi, Dr. Neel Shimpi, Dr. Arjun Singh, Dr. Mohita Sinha, Deeti Tarsaria, Dr. Herninder Kaur Thind, Gnana Kartheek Tirumalasetti, Eptehal Nashnoush and Professor Karmen Williams for their invaluable support in reviewing the papers. Their feedback and insights have significantly enriched the quality of this work.

## Author contributions

**Conceptualization:** Shrey Lakhotia, Leo Anthony Celi.

**Data curation:** Shrey Lakhotia, Chaitanya Sai Nutakki.

**Formal analysis:** Shrey Lakhotia.

**Investigation:** Shrey Lakhotia, Hormazd Godrej, Amandeep Kaur, Chaitanya Sai Nutakki, Michelle Mun, Pascal Eber.

**Methodology:** Shrey Lakhotia, Hormazd Godrej.

**Project administration:** Shrey Lakhotia.

**Resources:** Shrey Lakhotia, Hormazd Godrej, Chaitanya Sai Nutakki, Michelle Mun, Pascal Eber, Leo Anthony Celi.

**Software:** Shrey Lakhotia.

**Supervision:** Leo Anthony Celi.

**Validation:** Shrey Lakhotia.

**Visualization:** Shrey Lakhotia.

**Writing – original draft:** Shrey Lakhotia.

**Writing – review & editing:** Shrey Lakhotia, Amandeep Kaur, Leo Anthony Celi.

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
