## [Decision Letter · Decision Letter 0]

15 Apr 2025

PDIG-D-24-00600Machine learning in dentistry: a scoping reviewPLOS Digital Health Dear Dr. Celi, Thank you for submitting your manuscript to PLOS Digital Health. After careful consideration, we feel that it has merit but does not fully meet PLOS Digital Health's publication criteria as it currently stands. Therefore, we invite you to submit a revised version of the manuscript that addresses the points raised during the review process. Please submit your revised manuscript within 30 days. If you will need more time than this to complete your revisions, please reply to this message or contact the journal office at digitalhealth@plos.org. Please include the following items when submitting your revised manuscript: * A rebuttal letter that responds to each point raised by the editor and reviewer(s). You should upload this letter as a separate file labeled 'Response to Reviewers'. This file does not need to include responses to any formatting updates and technical items listed in the 'Journal Requirements' section below.* A marked-up copy of your manuscript that highlights changes made to the original version. You should upload this as a separate file labeled 'Revised Manuscript with Track Changes'.* An unmarked version of your revised paper without tracked changes. You should upload this as a separate file labeled 'Manuscript'. If you would like to make changes to your financial disclosure, competing interests statement, or data availability statement, please make these updates within the submission form at the time of resubmission. Guidelines for resubmitting your figure files are available below the reviewer comments at the end of this letter. We look forward to receiving your revised manuscript.Kind regards,Erika OngAcademic EditorPLOS Digital Health Erika OngAcademic EditorPLOS Digital Health Leo Anthony CeliEditor-in-ChiefPLOS Digital Healthorcid.org/0000-0001-6712-6626**Journal Requirements:**

1. We ask that a manuscript source file is provided at Revision. Please upload your manuscript file as a .doc, .docx, .rtf or .tex.

**Additional Editor Comments (if provided):****Reviewers' Comments:**Reviewer's Responses to Questions

**Comments to the Author**

1. Does this manuscript meet PLOS Digital Health’s publication criteria? Is the manuscript technically sound, and do the data support the conclusions? The manuscript must describe methodologically and ethically rigorous research with conclusions that are appropriately drawn based on the data presented.

Reviewer #1: Yes

Reviewer #2: Partly

Reviewer #3: Yes

Reviewer #4: Yes

2. Has the statistical analysis been performed appropriately and rigorously?

Reviewer #1: N/A

Reviewer #2: No

Reviewer #3: N/A

Reviewer #4: Yes

3. Have the authors made all data underlying the findings in their manuscript fully available (please refer to the Data Availability Statement at the start of the manuscript PDF file)?

Reviewer #1: Yes

Reviewer #2: Yes

Reviewer #3: Yes

Reviewer #4: Yes

4. Is the manuscript presented in an intelligible fashion and written in standard English?

Reviewer #1: Yes

Reviewer #2: Yes

Reviewer #3: Yes

Reviewer #4: No

5. Review Comments to the Author

Reviewer #1: This scoping review provides a much-needed overview of the landscape of machine learning in dentistry. It sheds light on important shortcomings in current research (notably the lack of calibration reporting, bias considerations, and open data). The paper is generally well-organized and informative. I offer the following constructive suggestions to further enhance the clarity and impact of the review:

• The Introduction could be made more concise, for example, the sentence “Dentistry... is ready to gain a lot from these advancements” could be rephrased in a more scholarly tone. Additionally, it’s better to avoid repetition – the phrase “to the best of our knowledge” appears twice in proximity.

• The Authors mention that previous reviews have explored ML in dentistry but that this work is unique in emphasizing explainability of metrics, calibration, and bias (which is an excellent positioning). It would strengthen the introduction to briefly cite those prior reviews and explicitly state how this review differs from them.

• The methodology is generally well-detailed and rigorous. One concern is the restriction to PubMed-indexed articles. Considering the interdisciplinary nature of ML, relevant studies might appear in computer science venues or other databases. In the manuscript’s Limitations, The authors acknowledge that limiting to PubMed and English could exclude some studies. To preempt readers’ concerns, it would help to briefly justify this in the Methods as well – for example, stating that PubMed was chosen for its coverage of biomedical journals.

• The authors should perhaps specify whether the multi-investigator reviewing process applied to full-text screening as well (it’s currently implied).

• One suggestion is to clarify in the Methods how the TRIPOD+AI rubric was applied. TRIPOD+AI is a reporting guideline; did the authors score each study on certain checklist items, or simply used it as an informal guide to decide which information to record (like calibration, interpretability, bias)?

• One area the authors might want to expand on is model validation approaches. Many readers will wonder: how many studies performed external validation or multi-center evaluation versus those that only did internal validation?

• Table 1 succinctly summarizes key performance metric reporting frequencies, which is a great addition. Depending on journal space, the authors might consider an additional figure or table to enrich the results. For example, a bar chart of the number of studies per dental specialty or per year could visually illustrate trends.

• To further enrich the discussion, the authors can consider adding concrete examples or referencing known techniques. For instance, when discussing bias and fairness, they can mention subgroup analysis and fairness-aware algorithms as remedies– citing a relevant study or framework (perhaps outside dentistry) that successfully applied such techniques, to illustrate what future dental ML studies could do.

• Since this is a scoping review, it would be helpful to explicitly state what the implications are for different stakeholders. The authors can consider mentioning implications for practitioners or educators.

• One additional future direction to mention could be the development or adoption of community data repositories or benchmarks for dental AI. Given that small number of studies shared their data publicly, the field would benefit from common datasets.

• Minor points: Overall, the manuscript is well-written. There are a few minor typographical and grammatical errors to correct for a polished final version. For example, “knolwedge” should be “knowledge”, and “simulatenously” should be “simultaneously”.

Reviewer #2: This manuscript addresses an important and rapidly evolving topic—the use of machine learning (ML) in dentistry—and commendably covers a wide breadth of studies. While the review is comprehensive in scope, incorporating the following refinements could further enhance clarity, transparency, and methodological rigor:

1. While scoping reviews can serve exploratory purposes, transparency and reproducibility would be further strengthened if a protocol is registered or published a priori (e.g., on Open Science Framework (OSF) or in journals that accept scoping review protocols). Just as a note for future scoping reviews: PROSPERO does not accept scoping review protocols.

2. Although PRISMA-ScR was used for reporting, it is unclear what methodological framework guided the conduct of the review (e.g., JBI methodology guidance for scoping reviews, Arksey and O'Malley, or its updated versions). Clarifying this could provide readers with greater insight into the review process.

3. Including the study types (study design) in the eligibility criteria could assist readers in understanding the weight and diversity of the evidence reviewed. Additionally, describing the study types in the "Characteristics of Included Studies" section would provide a clearer overview of the included literature.

4. Is there a specific rationale for selecting the time frame 2018 to 2023? If so, it would be helpful to mention this explicitly in the eligibility criteria.

5. Since the PICO framework is used, consider integrating it directly into the inclusion criteria section. This could streamline the methodology and avoid redundancy.

6. The PRISMA flow diagram could be further improved by including details such as the number of duplicates removed, specific reasons for article exclusion, and information at each screening stage.

7. The PRISMA-ScR checklist item “Synthesis of Results” is listed as reported on Page 4, but there is no clear description of how results will be synthesized on that page. This section could be a suitable place to describe the planned analyses, such as subgroup analysis or other methods of result synthesis.

8. It may be helpful to provide a comprehensive description of the characteristics of included studies (e.g., study design, geographical distribution) and use a thematic approach to present the findings in a more concise and organized manner, which could improve readability.

9. You highlighted the need to address equity and bias; however, providing specific examples from the reviewed literature could better demonstrate potential real-world implications or consequences of unaddressed biases.

10. Including practical recommendations for how dental practitioners can critically evaluate ML models before integrating them into clinical practice could further enhance the applicability of your findings.

11. While you acknowledged the limitation regarding the exclusive use of PubMed, it may be worth explicitly recommending that future reviews incorporate additional databases (e.g., Embase, Scopus) to ensure a more comprehensive literature search.

12. Minor Comments and Corrections:

• Grammar and Proofreading: Page 3, line 24: typo correction — "knolwedge" → "knowledge".

• Percentage Reporting: Ensure consistency in reporting percentages (e.g., use either 60% or 59.8% consistently).

• Figures: Consider including more graphical summaries (e.g., visual representations of dental specialties and ML tasks) to facilitate quick interpretation for readers.

Reviewer #3: The authors perform a scoping review of the dentistry literature to assess ML models in dentistry from 2018 to 2023. They follow the PICO framework and using PRISMA-ScR.

The overall quality of the review is high with careful attention paid to critical details regarding potential interpretability and downstream translation of these models. The emphasis on calibration and bias of the models is particularly important given the impact these elements will have on downstream translation. The search strategy only included PubMed and did not include IEEE Xplore for more technical type papers which may have offered greater methodological detail in modeling strategies. The authors should note the use of only one database in their limitations.

It would be helpful also to understand how many studies were in each type of application, for example, computer vision vs. tabular analysis to achieve its relevant tasks of classification, segmentation, etc.

Minor:

In the manuscript there is use of informal English such as "can't" rather than cannot and "couldn't" rather than could not. Would defer to the publisher regarding appropriate use of contractions.

Reviewer #4: General Assessment:

The manuscript explores a timely and relevant topic with well-defined objectives and a structured methodology, appropriate for a scoping review. However, several critical issues must be addressed before it can be considered for publication.

Major Concerns:

Language and Style:

The manuscript requires thorough language editing. Grammatical errors and awkward phrasing affect clarity. Redundancies (e.g., “To the best of our knowledge” appearing twice) and placeholders (e.g., "REFERENCE") should be corrected.

Methods:

The definition of machine learning is unclear. For example, exclusion criterion #1 may confuse traditional statistical methods with ML. The description of evaluation criteria lacks details on scoring, instruments used, and reviewer agreement resolution. The absence of PROSPERO registration needs clearer justification.

Results Interpretation:

Although percentages are presented clearly, the discussion is often superficial. Important findings, like the low reporting of calibration metrics, should be connected to their clinical relevance. Repetitive phrases should be consolidated, and vague language replaced with more precise analysis.

Bias, Equity, and Reproducibility:

The treatment of fairness and bias is shallow. Mentioning that 25% of studies addressed bias is not enough—examples and tools (e.g., AIF360, Fairlearn) should be discussed. Similarly, the low data sharing rate (11.72%) demands stronger commentary on its impact on reproducibility and open science.

Discussion and Conclusion:

The discussion is generally well-structured, with relevant ethical and regulatory context. However, claims about unsupervised learning trends contradict the data and should be revised. The discussion on interpretability needs concrete examples. Limitations should be more deeply analyzed, especially regarding the exclusive use of PubMed. The conclusion lacks a strong final message and should include 2–3 practical recommendations for future research.

6. PLOS authors have the option to publish the peer review history of their article (what does this mean?). If published, this will include your full peer review and any attached files.

**Do you want your identity to be public for this peer review?** For information about this choice, including consent withdrawal, please see our Privacy Policy.

Reviewer #1: No

Reviewer #2: No

Reviewer #3: No

Reviewer #4: No

**Figure resubmission:** While revising your submission, please upload your figure files to the Preflight Analysis and Conversion Engine (PACE) digital diagnostic tool, https://pacev2.apexcovantage.com/. PACE helps ensure that figures meet PLOS requirements. To use PACE, you must first register as a user. Registration is free. Then, login and navigate to the UPLOAD tab, where you will find detailed instructions on how to use the tool. If you encounter any issues or have any questions when using PACE, please email PLOS at figures@plos.org. Please note that Supporting Information files do not need this step. If there are other versions of figure files still present in your submission file inventory at resubmission, please replace them with the PACE-processed versions.**Reproducibility:**To enhance the reproducibility of your results, we recommend that authors of applicable studies deposit laboratory protocols in protocols.io, where a protocol can be assigned its own identifier (DOI) such that it can be cited independently in the future. Additionally, PLOS ONE offers an option to publish peer-reviewed clinical study protocols. Read more information on sharing protocols at https://plos.org/protocols?utm_medium=editorial-email&utm_source=authorletters&utm_campaign=protocols

---

## [Decision Letter · Decision Letter 1]

25 Jun 2025

Machine learning in dentistry: a scoping review

PDIG-D-24-00600R1

Dear Dr Celi,

We are pleased to inform you that your manuscript 'Machine learning in dentistry: a scoping review' has been provisionally accepted for publication in PLOS Digital Health.

Best regards,

Erika Ong

Academic Editor

PLOS Digital Health

**Additional Editor Comments (if provided):**

**Reviewer Comments (if any, and for reference):**

Reviewer's Responses to Questions

**Comments to the Author**

1. If the authors have adequately addressed your comments raised in a previous round of review and you feel that this manuscript is now acceptable for publication, you may indicate that here to bypass the “Comments to the Author” section, enter your conflict of interest statement in the “Confidential to Editor” section, and submit your "Accept" recommendation.

Reviewer #1: All comments have been addressed

Reviewer #2: All comments have been addressed

Reviewer #3: All comments have been addressed

Reviewer #4: (No Response)

2. Does this manuscript meet PLOS Digital Health’s publication criteria? Is the manuscript technically sound, and do the data support the conclusions? The manuscript must describe methodologically and ethically rigorous research with conclusions that are appropriately drawn based on the data presented.

Reviewer #1: Yes

Reviewer #2: Yes

Reviewer #3: Yes

Reviewer #4: Partly

3. Has the statistical analysis been performed appropriately and rigorously?

Reviewer #1: N/A

Reviewer #2: Yes

Reviewer #3: N/A

Reviewer #4: No

4. Have the authors made all data underlying the findings in their manuscript fully available (please refer to the Data Availability Statement at the start of the manuscript PDF file)?

Reviewer #1: Yes

Reviewer #2: (No Response)

Reviewer #3: Yes

Reviewer #4: Yes

5. Is the manuscript presented in an intelligible fashion and written in standard English?

Reviewer #1: Yes

Reviewer #2: (No Response)

Reviewer #3: Yes

Reviewer #4: No

6. Review Comments to the Author

Reviewer #1: I appreciate the time and effort that the authors have put into correcting the original manuscript and incorporating the feedback.

I agree with the changes made and have no additional comments.

Reviewer #2: (No Response)

Reviewer #3: (No Response)

Reviewer #4: Thank you for the opportunity to review your manuscript entitled “Machine learning in dentistry: a scoping review.” This work addresses a timely and important topic at the intersection of artificial intelligence and dental clinical practice. The study is methodologically grounded, follows PRISMA-ScR guidelines, and provides a broad overview of 280 studies published between 2018 and 2023.

While the manuscript has strong potential, it does not yet meet the full publication criteria of PLOS Digital Health in its current form. Below are detailed comments and recommendations for improvement:

1. Language and Presentation

- The manuscript requires significant language editing. The English is often unclear or inconsistent, with a mix of academic and informal tone. Phrases like “dentistry has a long way to go” and “this is critical, as…” weaken scientific precision.

- Some terminology is used without clear definition (e.g., “generative tasks,” “direct point-of-care use”), and terms like “bias” and “fairness” are often interchanged without technical clarity.

Recommendation: Have the manuscript professionally edited for clarity, consistency, and scientific tone.

2. Redundancy and Structure

- The manuscript frequently repeats concepts across sections, particularly related to calibration, explainability, and bias. This weakens the impact of the key messages.

- The Introduction and Discussion sections contain overlapping content that can be streamlined.

Recommendation: Consolidate redundant sections and focus each paragraph on a distinct, well-supported point.

3. Analytical Depth

- The analysis is largely descriptive (e.g., percentage reporting of various model features), which is appropriate for a scoping review. However, the conclusions drawn often overreach the data, suggesting systemic field-wide risks without deeper statistical or stratified analysis.

- Claims regarding calibration, fairness, and risk in real-world implementation would benefit from more nuanced analysis — for example, by grouping results by geography, algorithm type, or model task.

Recommendation: Limit conclusions to what is strictly supported by the data, or enhance the analysis to justify broader claims.

4. Transparency and Supplementary Material

- The manuscript refers to supplementary materials (e.g., S2 File with extracted data), which is good practice. However, it was not possible to assess these files from the PDF itself.

Recommendation: Ensure all supporting files (search strategy, extracted data) are included and clearly referenced in submission.

5. Ethical and Methodological Standards

- The manuscript meets ethical requirements. It does not involve human subjects or private data, and the methodology is aligned with scoping review standards.

- The use of TRIPOD+AI criteria is a strength and improves reporting rigor.

Overall Recommendation: Minor Revision

The topic is highly relevant, and the paper is methodologically promising. However, the manuscript requires substantial improvements in:

- Language and clarity;

- Depth and precision of analysis;

- Alignment between results and conclusions;

- Overall scientific tone and structure.

Addressing these points will significantly improve the manuscript’s clarity, impact, and suitability for publication.

7. PLOS authors have the option to publish the peer review history of their article (what does this mean?). If published, this will include your full peer review and any attached files.

**Do you want your identity to be public for this peer review?** For information about this choice, including consent withdrawal, please see our Privacy Policy.

Reviewer #1: No

Reviewer #2: No

Reviewer #3: No

Reviewer #4: No
